# Injury Prevalence among Young Elite Baseball Players

**DOI:** 10.3390/sports11070134

**Published:** 2023-07-17

**Authors:** Daeho Ha, Satoshi Nagai, Byungjoo Noh, Naoki Mukai, Shumpei Miyakawa, Masahiro Takemura

**Affiliations:** 1Department of Sports Medicine, Graduate School of Comprehensive Human Sciences, University of Tsukuba, 1-1-1 Tennodai, Tsukuba 305-8577, Japan; ha.daeho@gmail.com; 2Department of Physical Therapy, Faculty of Health Sciences, Tsukuba International University, 6-8-33 Manabe, Tsuchiura 300-0051, Japan; s-nagai@tius.ac.jp; 3Department of Kinesiology, Jeju National University, Jeju 63243, Republic of Korea; bnoh@jejunu.ac.kr; 4Faculty of Health and Sport Sciences, University of Tsukuba, 1-1-1 Tennodai, Tsukuba 305-8577, Japan; mukai.naoki.fu@u.tsukuba.ac.jp; 5Faculty of Medicine, University of Tsukuba, 1-1-1 Tennodai, Tsukuba 305-8577, Japan; miyakawa.shumpei.fn@u.tsukuba.ac.jp

**Keywords:** epidemiology, baseball injury, young baseball players, injury prevalence, early single sport specialization

## Abstract

This study aimed to describe the injury profiles of young Korean baseball players according to position and age as the proportion and distribution of injuries based on playing position and age remains unclear. A total of 271 elite youth baseball players aged 8 to 16 years were divided into two groups: elementary school (ES) (n = 135) and middle school (MS) (n = 136). The participants’ basic, baseball practice, and injury information were collected. Injuries in the MS group were not limited to the elbow and shoulder, and injury prevalence varied by age group and baseball position. The most injured body region in the ES group was the elbow joint, regardless of the position. In contrast, the most injured body region in the MS group was the lower back, except for infielders whose elbows were the most injured. Additionally, the MS group was more likely to experience injuries to the lower back (OR = 4.27, 95% CI = 2.47–7.40), shoulder (OR = 1.93, 95% CI = 1.08–3.43; *p* = 0.024), and knee (OR = 2.15, 95% CI = 1.17–3.94; *p* = 0.012). Our findings indicate that excessive practice and a lack of rest during MS (growth spurt period) can significantly increase the risk of lower back problems in young baseball players.

## 1. Introduction

Baseball is a common sport in which early sports specialization begins in young adolescents [1,2]. Currently, early sports specialization in youth and overuse injuries continues to increase in frequency [3,4]. Young baseball players are more susceptible to overuse injuries due to a variety of factors, such as skeletal immaturity, poor biomechanics, and heavy workloads in elite sports [5,6]. During the middle school years (players are aged 12–15 years), when bones and joints rapidly undergo substantial growth, there is a risk of injury owing to excessive exercise and poor movement [7]. Lyman et al. [8] reported that serious pitching injuries are most likely caused by cumulative trauma that begins in childhood. Kraut et al. [9] also reported that students with year-round youth baseball activities have an increasing incidence of pitching-related injuries, which are evident in high school and college, but originate during earlier years. However, few studies have reported injury distribution by age group (elementary and middle school).

Injuries to the upper extremities are more prevalent in pitchers, while injuries to the lower extremities, particularly the knee joints, are more prevalent in fielders [8]. Upper-extremity injuries are more common in pitchers, while fielders tend to suffer from lower extremity injuries, especially the knee joints [8]. Numerous studies have been conducted on injuries in young baseball players, most of which have focused on injuries to the elbow and shoulder joints [10,11,12,13,14,15,16,17]. However, many injuries also occur in other body parts during the growth period. Only a few studies have investigated injuries in different body regions [3,18,19]. Moreover, trunk and lower limb injuries, such as spondylolysis [20,21], Osgood-Schlatter disease [22,23], and Sever’s disease [22,23], frequently occur during the growth period. Lower limb and trunk injuries can also lead to shoulder and elbow injuries [24]. Investigating whole-body injuries may help prevent shoulder and elbow injuries in baseball players during the growth period. Therefore, it is necessary to investigate the association between the incidence of injury sites and baseball positions in young baseball players.

It is known that injury risk increases with age and competition level [25]. In South Korea, young baseball players are divided into varsity teams with an elite team and a non-elite team. The elite-level team (in the school league) practices more than the non-elite team (in the club league) [26]. Although they undergo vigorous training and compete seriously, their injuries remain unknown. This study aimed to describe the injury profiles of elite Korean youth baseball players according to their baseball positions and age groups.

## 2. Materials and Methods

### 2.1. Participants

A total of 271 elite male youth Korean baseball players ranging in age from 8 to 16 years were included. They were divided into two groups: elementary school (ES), comprising individuals aged 8–13 years, and middle school (MS), comprising individuals aged 12–16 years. The ES group consisted of 135 male baseball players (age 10.9 ± 1.0 years; height 148.3 ± 8.5 cm; weight 45.6 ± 10.3 kg; baseball career 2.2 ± 1.6 years). The MS consisted of 136 male baseball players (age 14.0 ± 1.1 years; height 166.9 ± 9.1 cm; weight 64.5 ± 12.0 kg; baseball career 4.2 ± 1.5 years) (Table 1). The study was conducted with consent obtained from all elite baseball players, coaches, and parents, and was approved by the University of Tsukuba Research Ethics Committee (approval number: 28–64).

### 2.2. Questionnaire

The questionnaire was distributed and completed during the off-season from September to October 2015. After obtaining permission to participate, the researchers visited each school and conducted a survey. As all the participants were underage, the research team explained the research objective and questionnaire items to help them understand. All participants completed all items and submitted them. The questionnaire captured the following information:(1)Basic information including age (date of birth), grade, weight, height, baseball career, main position, throwing side, and batting side.(2)Baseball practice frequency per week and practice time per day (team and individual).(3)Information of previous and current injury body region, injury situation, date of injury, hospital treatment or not, diagnosis name of the injury, surgery or not, and period of recovery from injury.

### 2.3. Injury Definition

Referring to previous studies, injury was defined as a restriction from participating in baseball activities (practice and games) for more than one day due to an incident that occurred while participating in baseball activities [27,28].

### 2.4. Statistical Analysis

Descriptive data were obtained using a Microsoft Excel spreadsheet that removed any personal information. The injury data were classified into eight body regions based on the Major League Baseball’s Health and Injury Tracking System [29] and Orchard Sports Injury Classification System [30]. Statistical analysis was carried out using SPSS 26.0 (IBM SPSS Statistics for Windows, IBM Corp., Armonk, NY, USA). We compared the physical characteristics (age) of the ES and MS groups using independent t-tests. We examined the frequency of injury to the body according to the age group and baseball position using multiple-response crosstabs. We used Pearson’s chi-square tests to determine differences between the ES and MS groups according to the injured body region. In all cases, the significance level was set at *p* < 0.05.

## 3. Results

### 3.1. Subject’s Injury Prevalence with Practice Time and Frequency by Each Age Group

The injury prevalence of the participants showed significant differences between the groups (Table 2). Among all participants, 82.7% (n = 224) had an injury experience while 17.3% (n = 47) had no injuries. The proportion of injury to non-injury patients was 75.6% (n = 102) to 24.4% (n = 33), respectively, in the ES age group, and 89.7% (n = 122) to 10.3% (n = 14), respectively, in the MS age group. The MS group reported a significantly longer practice time (*p* < 0.001) and lower practice frequency (*p* = 0.035).

### 3.2. Total Frequency of Injury in Each Baseball Position

Pitchers showed high injury prevalence in the elbow (49.2%), lower back (44.2%), and shoulder (33.3%). Almost half of the patients experienced injuries, especially to the elbow and lower back. There were 113 total injuries among the 63 pitchers. Catchers experienced high injury prevalence in the lower back (41.7%), elbow (37.5%), and shoulder (33.3%). The total number of injury events was 42 for the 24 catchers. Outfielders showed high injury prevalence in the lower back (31.4%), elbow (30.0%), and shoulder (25.7%). They had higher injury prevalence in the ankle (22.9%) and hand (18.9%) than the pitchers and catchers. The total number of injury events was 119 in 70 outfielders. Infielders showed a high prevalence of injuries in the elbow (34.9%), lower back (26.4%), and knee (21.7%). Shoulder injury prevalence (15.1%) in infielders was higher than the other three position groups. Likewise, outfielders had higher injury percentages for the ankle (17.9%) and hand (16%) than pitchers and catchers. The total number of injuries was 158 among the 106 infielders (Table 3).

### 3.3. ES Group Frequency of Injury in Each Baseball Position

Across all the positions, the ES group had the highest prevalence of elbow injury (Table 3). Injury prevalence in the pitcher position was the highest in the elbow (65%), lower back (20%), and foot (20%), with 30 injuries occurring in 20 pitchers. Catchers experienced frequent injuries to the elbow (41.7%), foot (33.3%), and shoulder (25%), with 16 injuries reported in 12 catchers. Infielders showed a high prevalence of injuries in the elbow (30.6%), lower back (21%), and ankle (17.7%), with 79 injury events in 62 infielders. Outfielders displayed a high prevalence of injuries in the elbow (24.4%), shoulder (24.4%), and knee (22%), with 56 injury events in 41 outfielders.

### 3.4. Frequency of Injury in Each Baseball Position among the MS Group

For pitchers, catchers, and outfielders in the MS group, the prevalence of lower back injuries was the highest, whereas for infielders, the prevalence of elbow injuries was higher (Table 3). In the pitcher position among MS players, the most common injuries were the lower back (55.8%), shoulder (44.2%), and elbow (41.9%), with 83 injuries in 43 pitchers. Catchers experienced a high prevalence of injuries in the lower back (75%), shoulder (41.7%), and elbow (33.3%), with 26 injury events in 12 participants. Infielders showed a high prevalence of injuries in the elbow (40.9%), knee (36.4%), and lower back (34.1%), with 79 injury events in 44 participants. Outfielders showed frequent injuries in the lower back (51.7%), elbow (37.9%), and ankle (31.0%), with 63 injury events occurring in 29 outfielders.

### 3.5. Injury Frequency and Injury Odds Ratio for Each Body Region of the MS Group Compared to the ES Group

The ES group reported a high prevalence of injuries in the elbow (34.8%), lower back (18.5%), shoulder (17.8%), and feet (17.0%). The MS group reported a high prevalence of injuries in the lower back (49.3%), elbow (39.7%), shoulder (29.4%), and knee (27.2%). The MS group had a 4.27 times higher potential prevalence of a lower back injury (OR = 4.27, 95% CI = 2.47–7.40) than the ES group, indicating a significant difference (*p* = 0.000). The MS group also indicated a 1.93 times higher potential prevalence of a shoulder injury (OR = 1.93, 95% CI = 1.08–3.43; *p* = 0.024) and 2.15 times higher prevalence of a possible knee injury (OR = 2.15, 95% CI = 1.17–3.94; *p* = 0.012). By contrast, the MS group had a 0.35 times lower potential prevalence of a foot injury (OR = 0.35, 95% CI = 0.15–0.78; *p* = 0.008) than the ES group (Table 4).

## 4. Discussion

This study showed that the body regions most injured in elite Korean youth baseball players depended on their age group and baseball position. (1) The prevalence of injury was 75.6% and 89.7% among the ES and MS groups, respectively. (2) The MS players had more practice time per day and lower practice frequency than the ES players. (3) ES players suffered mostly elbow injuries, regardless of their position, while MS players mostly experienced injuries to the lower back, except for infielders who suffered mostly elbow injuries. (4) Compared to the ES group, the MS group had more lower back, shoulder, and knee injuries and fewer foot injuries.

### 4.1. Injury Prevalence Differed between Age Groups with Practice Time and Frequency

The practice time and frequency were significantly different between the two groups: 6.7 h and 6.5 days, respectively, in the ES group, and 7.8 h and 6.2 days, respectively, in the MS group. The total injury prevalence among the subjects was 82.7%, 75.6%, and 89.7% among the ES and MS groups, respectively. This high prevalence is due to excessive practice—the Republic of Korea has an excessive amount of practice compared to the United States [31,32], Japan [16,33], and Europe [34]. The subjects of this research also began participating in the elite program system at a young age when competition was intense, and players rarely took breaks during the week. The high incidence of injury in young baseball players is assumed to be a consequence of these complicated circumstances. Therefore, our results provide valuable reference materials for preventing future injuries in young baseball players.

### 4.2. Injury Prevalence by Body Regions and Positions

We compared the distribution of injury sites according to the age of the young players and the distribution of injury sites by baseball position in the ES and MS groups. In the ES group, elbow injury was the most prevalent, regardless of the position. Only outfielders reported an elbow injury at the same rate as a shoulder injury. The prevalence rates of elbow and shoulder injury were 65.0% and 10.0%, respectively, for pitchers; 41.7% and 25.0%, respectively, for catchers; and 30.6% and 14.5%, respectively, for infielders. The prevalence of elbow injury was particularly high among pitchers and catchers. These findings suggest that young baseball players, especially elementary school players, are most susceptible to elbow injuries. Matsuura et al. [17] reported that elbow injury was associated with age, pitcher, and catcher positions, and longer training hours per week. Because pitchers and catchers pitch and throw more than players in other positions, owing to the characteristics of their positions, the incidence of elbow injuries is high [11,35]. In addition, the positions of pitchers and catchers are known risk factors for elbow injuries [12,17,36]. A competition system at the elite school level significantly affects advancement to middle and high schools, which could lead to overuse injuries in pitchers. Our results also showed that pitchers in the ES group had a six-fold higher prevalence of elbow injuries than shoulder injuries. In addition, our results imply that they continue to pitch with a fatigued arm because of excessive practice time and a lack of rest. Therefore, we recommend that coaches concentrate on avoiding and managing elbow injuries by comprehensively monitoring pitchers and catchers.

The MS group experienced the most lower back injuries in all positions except for the infielder. Our results suggest that middle school-aged elite youth baseball players may be vulnerable to lower back injuries if they practice excessively and without recovery time. However, our results were inconsistent with the order of prevalence by position among baseball players reported in previous studies [37]. Movements such as the repetition of rotation to one side during pitching (pitchers) and throwing (catchers), one-side rotation and swing amount (including an empty swing) during the swing, flexion of the upper body during defense (infielders), and instantaneous running (outfielders) could be considered as mechanisms of lower back injury in the peak height velocity period [38]. According to the reports by Ferguson et al. [5] and Zaremski et al. [6], lower back injury increases during the growth phase due to several factors such as the immaturity of the bone structure and a lack of adequate biomechanics. Furthermore, there are large differences in the frequency of spondylolysis among athletes who play certain sports. Youth baseball players may be more prone to injuries if their sports involve a lot of bending and straightening of the lumbar spine, especially when combined with rotation [39]. Although it is not investigated in this study, it is estimated that elite Korean youth baseball players have experienced or are much more likely to experience stress fractures. Therefore, a further investigation of spondylolysis is necessary.

### 4.3. Injury Prevalence by Body Regions of MS Group Based on ES Group

The MS group had more lower back, shoulder, and knee injuries, but fewer foot injuries than the ES group. The MS group had 30.8% higher prevalence than the ES group (49.3% vs. 18.5%). MS players were 4.27 times more likely than ES players to experience lower back injury (OR = 4.27, 95% CI = 2.47–7.40). It is known that the frequency of lower back injury in youth baseball players (aged 12–15.5 years) is between 8.5% and 14.6% of youth baseball players; however, the MS group showed a prevalence that is approximately 38% higher than that reported in previous studies [40,41,42]. In contrast, the ES group had an average practice time of 6.7 h per day, which was not significantly higher than that recorded in a previous study [43]. In addition, the prevalence of lower back injury was 18.5%, which is higher than the previously reported prevalence rate [24]. This is due to the accumulation of such overuse since elementary school; therefore, the prevalence of lower back injury seems to have increased in middle school students. Thus, it is necessary to manage the practice time, frequency, and games from elementary school to prevent a massive increase in lower back injuries during middle school.

The MS group had a higher incidence of lower back injury (49.3%) than the elbow (39.7%) and shoulder (29.4 %) injury groups. Furthermore, the MS group was 4.27 times more likely to experience lower back injury than the ES group (OR = 4.27, 95% CI = 2.47–7.40). The elbow and shoulder joints were the most frequently injured body regions in middle school baseball players. However, the results of the present study differ from those of previous studies. According to d’Hemecourt et al. [43], improper techniques and overtraining can lead to back problems, particularly during the rapid growth phase, such as the peak height velocity period. In addition, the practice volume and intensity of suitable training for young athletes differs among players [44]. The required practice volume and intensity may change as young players develop and mature. However, a combination of circumstances, such as playing at an elite level, skeletal immaturity during rapid growth, a lack of adequate biomechanics, and a severe workload increases the risk of overuse injuries in youth baseball players [5,6]. This study showed that the average practice time per day in the MS group was a staggering 7.8 h, which could lead to significant lower back injury and spinal issues, even though they required personal management according to growth velocity. Another issue is that elbow and shoulder injuries in young baseball players are associated with the disruption of the kinetic chain caused by lower back injuries [10]. Our results showed that the MS group players were more likely to experience elbow (39.7%) and shoulder injury (29.4%) than the ES group players (34.8% and 17.8%, respectively). The data in this study did not include data for each year for individual follow-ups; nonetheless, elbow and shoulder injuries can also be a significant concern for elite young athletes undergoing rapid growth. Thus, the monitoring and management of shoulder and elbow injuries are required to prevent back pain and spinal injuries in elite youth baseball players. Further studies on the relationship between lower back and shoulder injuries are required.

Players in the MS group were more likely to experience a lower extremity injury (knee injury, 27.2%; foot injury, 6.6%) than those in the ES group. The prevalence of lower back and knee injuries among young baseball players (6–15 years old) was 8.4% and 13.1%, respectively. A previous study reported that knee injury is associated with lower back injury. Using the absence of knee injury as a reference, the adjusted odds ratio for a lower back injury was 5.83 in the presence of a knee injury [45]. Thus, a knee injury is more likely to occur in the acute period along with a lower back injury. In addition, young baseball players with a knee injury are more likely to have a lower back injury. The prevalence of foot injury was 10.4% lower in the MS group than in the ES group. This result can be seen as the result of memory bias in the MS group rather than the reason that the subjects differed. From this memory bias in the MS group players, we can infer that foot injury is not a significant problem for youth baseball players during the MS period.

This study had several limitations. First, it was a retrospective study based on questionnaires, and there may be memory bias. Second, the volume and frequency of practice per week and the extent to which the lack of rest affected injuries were not assessed. Third, classification according to the injury type and severity was not performed (the inability to play baseball activity due to pain or injury for more than one day was counted as an injury). Fourth, as our study selected only elite-level youth baseball players, a direct comparison with control groups, such as club and recreation levels, was not performed. Nevertheless, our study is of sufficient value as it is, to our knowledge, the first report on the injury status of elite Korean youth baseball players specializing in sports at an early age. This study provides significant evidence for future research on injury prevention.

## 5. Conclusions

In this study, we found that baseball players in elite youth programs had a higher risk of injury due to excessive practice and a lack of rest, and the frequency of injuries varied by position and age. ES players showed a high prevalence of elbow injuries in all baseball positions. For MS players in all positions (except infielders), lower back injuries were the most prevalent. In addition, MS players were significantly more likely to have lower back, shoulder, and knee injuries than ES players. Our results suggest the need to pay special attention to lower back injuries in baseball players during the MS period (peak growth period). In the future, through a prospective study, it is necessary to clarify the spinal injuries of elite baseball players in the growing age more precisely and clearly.

## Figures and Tables

**Table 1 sports-11-00134-t001:** Demographic characteristics of the participants.

	Total (n = 271)	ES (n = 135)	MS (n = 136)	*p*-Value
Age (years)	12.5 (1.9)	10.9 (1.0)	14.0 (1.1)	<0.001 *
Height (cm)	157.6 (12.8)	148.3 (8.5)	166.9 (9.1)	<0.001 *
Weight (kg)	55.1 (14.7)	45.6 (10.3)	64.5 (12.0)	<0.001 *
BMI (kg/m^2^)	21.8 (3.5)	20.6 (3.4)	23.0 (3.1)	<0.001 *
Baseball career (years)	3.2 (1.9)	2.2 (1.6)	4.2 (1.5)	<0.001 *

Mean (standard deviation); ES, elementary school; MS, middle school; BMI, body mass index; Asterisks denote a significant difference between groups (* *p* < 0.05).

**Table 2 sports-11-00134-t002:** Prevalence of injuries, practice time, and frequency by age group.

	Total (n = 271)	ES (n = 135)	MS (n = 136)	*p*-Value
Practice time (hours/day)	7.2 (1.4)	6.7 (1.0)	7.8 (1.6)	<0.001 *
Practice frequency (day/week)	6.4 (0.5)	6.5 (0.5)	6.2 (0.5)	0.035 *
Injury prevalence rate (n)	82.7% (224)	75.6% (102)	89.7% (122)	-
Non-injury prevalence rate (n)	17.3% (47)	24.4% (33)	10.3% (14)	-

Mean (standard deviation); ES, elementary school; MS, middle school; Asterisks denote a significant difference between groups (* *p* < 0.05).

**Table 3 sports-11-00134-t003:** Frequency of injury by body region, position, and age group.

Rank	Pitcher (n = 63)			Catcher (n = 24)			Infielder (n = 106)			Outfielder (n = 70)		
	Body Region	No.	%	Body Region	No.	%	Body Region	No.	%	Body Region	No.	%
Total		113			42			158			119	
1	Elbow	31	49.2	Lower back	10	41.7	Elbow	37	34.9	Lower back	22	31.4
2	Lower back	28	44.4	Elbow	9	37.5	Lower back	28	26.4	Elbow	21	30.0
3	Shoulder	21	33.3	Shoulder	8	33.3	Knee	23	21.7	Shoulder	18	25.7
4	Knee	13	20.6	Knee	4	16.7	Ankle	19	17.9	Ankle	16	22.9
5	Ankle	8	12.7	Foot	4	16.7	Hand	17	16.0	Knee	15	21.4
6	Foot	7	11.1	Wrist	3	12.5	Shoulder	16	15.1	Hand	13	18.6
7	Wrist	3	4.8	Ankle	2	8.3	Foot	12	11.3	Foot	9	12.9
8	Hand	2	3.2	Hand	2	8.3	Wrist	6	5.7	Wrist	5	7.1
ES		30			16			79			56	
1	Elbow	13	65.0	Elbow	5	41.7	Elbow	19	30.6	Elbow	10	24.4
2	Lower back	4	20.0	Foot	4	33.3	Lower back	13	21.0	Shoulder	10	24.4
3	Foot	4	20.0	Shoulder	3	25.0	Ankle	11	17.7	Knee	9	22.0
4	Knee	3	15.0	Wrist	2	16.7	Hand	10	16.1	Lower back	7	17.1
5	Shoulder	2	10.0	Lower back	1	8.3	Shoulder	9	14.5	Ankle	7	17.1
6	Ankle	2	10.0	Knee	1	8.3	Foot	9	14.5	Foot	6	14.6
7	Hand	2	10.0	Ankle	0	0.0	Knee	7	11.3	Hand	5	12.2
8	Wrist	0	0.0	Hand	0	0.0	Wrist	1	1.6	Wrist	2	4.9
MS		83			26			79			63	
1	Lower back	24	55.8	Lower back	9	75.0	Elbow	18	40.9	Lower back	15	51.7
2	Shoulder	19	44.2	Shoulder	5	41.7	Knee	16	36.4	Elbow	11	37.9
3	Elbow	18	41.9	Elbow	4	33.3	Lower back	15	34.1	Ankle	9	31.0
4	Knee	10	23.3	Knee	3	25.0	Ankle	8	18.2	Shoulder	8	27.6
5	Ankle	6	14.0	Ankle	2	16.7	Shoulder	7	15.9	Hand	8	27.6
6	Foot	3	7.0	Hand	2	16.7	Hand	7	15.9	Knee	6	20.7
7	Wrist	3	7.0	Wrist	1	8.3	Wrist	5	11.4	Foot	3	10.3
8	Hand	0	4.7	Foot	0	0.0	Foot	3	6.8	Wrist	3	10.3

Eight participants with no main position were excluded.

**Table 4 sports-11-00134-t004:** Injury frequency and injury odds ratio for each body region of MS Group compared to ES Group.

Body Region	Total		ES		MS		OR	95% CI	*p* Value
	No.	%	No.	%	No.	%			
Elbow	101	37.3	47	34.8	54	39.7	1.23 ^a^	0.75–2.02	0.405
Lower back	92	33.9	25	18.5	67	49.3	4.27 ^b^	2.467–7.399	0.000 *
Shoulder	64	23.6	24	17.8	40	29.4	1.93 ^c^	1.084–3.425	0.024 *
Knee	57	21.0	20	14.8	37	27.2	2.15 ^d^	1.171–3.942	0.012 *
Ankle	46	17.0	20	14.8	26	19.1	1.36 ^e^	0.717–2.575	0.345
Hand	35	12.9	17	12.6	18	13.2	1.06 ^f^	0.520–2.154	0.875
Foot	32	11.8	23	17.0	9	6.6	0.35 ^g^	0.153–0.777	0.008 *
Wrist	18	6.6	5	3.7	13	9.6	2.75 ^h^	0.952–7.935	0.053
Total	445		181		264				

ES, elementary school; MS, middle school; OR, odds ratio; CI, confidence interval; Asterisks denote ^a^ significant difference between groups (*p* < 0.05); a 0 cells (0.0%) have an expected event of less than 5. The minimum expected event is 50.31; ^b^ 0 cells (0.0%) have an expected event of less than 5. The minimum expected event is 45.83; ^c^ 0 cells (0.0%) have an expected event of less than 5. The minimum expected event is 31.88; ^d^ 0 cells (0.0%) have an expected event of less than 5. The minimum expected event is 28.39; ^e^ 0 cells (0.0%) have an expected event of less than 5. The minimum expected event is 22.92; ^f^ 0 cells (0.0%) have an expected event of less than 5. The minimum expected event is 17.44; ^g^ 0 cells (0.0%) have an expected event of less than 5. The minimum expected event is 15.94; ^h^ 0 cells (0.0%) have an expected event of less than 5. The minimum expected event is 8.97.

## Data Availability

The data presented in this study are available upon request from the corresponding author.

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
