# Peer review of "Injury Prevalence among Young Elite Baseball Players"

_sports, 2023, doi:10.3390/sports11070134_

Round 1

Reviewer 1 Report

Dear authors: The paper is well written. Here are some observations to improve your understanding. First of all, you indicate that you carried out the study between September and October 2015. Almost 8 years ago. Can you explain this delay? After so much time, it is very difficult for the protagonists to take advantage of their results and lose the social and sporting significance.

You indicate that they studied elite players and use (26) as a reference, but this quote is not accessible to Western readers. Can you explain how you ranked the kids into elite vs non-elite? and provide a reference that can be read.

Reference 8 is also not accessible. Please provide an updated one.

The results indicate that the participants trained an average of 7.2 (1.4) hours per day and even the participants in the MS group trained an average of 7.8 (1.6) h/d. They also trained 6.4 days a week. Can you confirm these data? Is this training schedule compatible with the school calendar? If so, indicate the training schedule from Monday to Friday, Saturdays and Sundays, as well as the time dedicated to training for general physical preparation, technical preparation and other sports. Indicate the number of months of training per year and your previous experience. Could you indicate the useful training time or the number of pitches, hits or runs per training? You can describe practical exposure data like Jayanthi (2015) to be able to make comparisons.

In the results they could indicate the relationship between practice exposure data (pitches, hits, runs) and specific injuries.

Thank you so much.

Reviewer 2 Report

To the authors:

In my opinion, the title is misleading as it talks about the injury frequency of "elite" baseball players.

In the thesis there is a reference to the classification of injury, but it seems that the distinction between pain and injury has not been defined. Spodilolysis, Schlatter disease, Sever disease are not considered classic injuries, but rather sports harm or age-related complaints.

Another problem is the content of the term "Elite". In international literature, elite means a higher quality, this is not typical at the age of 10 years.

The choice of sample (primary and secondary school) is an additional problem in the topic discussed.

Although there are several references to the load capacity of the young body, the group averages indicate that the musculoskeletal problems of the young group (10 years old) before puberty are of a different nature than those of 14-year-olds.

I recommend explaining the difference between pain, injury and harm in the thesis and, if possible, analyzing the sample from this point of view.

The value of the research is greatly enhanced by the fact that in both groups, the frequency rate of "injuries" is very high, 75.90% was measured by the authors. These frequencies draw attention to the problems of training. Unfortunately, the regular physical exertion that may be behind the observed problems is unknown.

I propose to correct the thesis as follows:

1. It is necessary to discuss the difference between pain and injury more precisely.

2. A one-day absence is not relevant in case of injury, it is a one-day injury, not an injury in the classical sense.

3. The alarmingly high number of cases makes it necessary to explain the causes and make suggestions for intervention. This would greatly increase the value of the research.

4. The possibility of the proposed corrections is possible, as the methodology described under point 3 of the query provides this information.

Reviewer 3 Report

This is an epidemiological study in young baseball players.

The study is interesting. However, there is a major flaw: which questionnaire has been used and has it been validated ?

So, to be published, the paper must report the questionnaire as a whole, citing the sources and the validation procedure.

Round 2

Reviewer 2 Report

To the authors:

The answers of the authors were accepted generaly, with some comments.

1.The title "Injury prevalence among young elite baseball players" may be correct in Korea, but not in an international context. I suggest: Injury prevalence among young basketball players age between......in Korea.

2. A working definition is required in the problem determination ofof injury, pain and desease, harm

Reviewer 3 Report

Publish after the methdological paper will be published, as the authors stated:

"

Revised: We agree with your opinion. Unfortunately, however, there remains an unpublished task using the questionnaire used in this study. If this is not rude, please continue to follow the papers that our research team presents in the future. Thank you so much.

Author Response

Publish after the methdological paper will be published, as the authors stated

Revised: Thank you for your comment. We accept that we need to report the questionnaire as you mentioned in Round 1 ("So, to be published, the paper must report the questionnaire as a whole, citing the sources and the validation procedure"). As there remains an unpublished task using the questionnaire used in this study, we attached appendix which contains the contents of the questionnaire only used in this study. Please check the attached document. 
